# Towards Skilled Population Curriculum for Multi-Agent Reinforcement Learning

## Abstract

Recent advances in multi-agent reinforcement learning (MARL) allow agents to coordinate their behaviors in complex environments. However, common MARL algorithms still suffer from scalability and sparse reward issues. One promising approach to resolving them is *automatic curriculum learning* (ACL). ACL involves a *student* (curriculum learner) training on tasks of increasing difficulty controlled by a *teacher* (curriculum generator). Despite its success, ACL's applicability is limited by (1) the lack of a general student framework for dealing with the varying number of agents across tasks and the sparse reward problem, and (2) the non-stationarity of the teacher's task due to ever-changing student strategies. As a remedy for ACL, we introduce a novel automatic curriculum learning framework, Skilled Population Curriculum (SPC), which adapts curriculum learning to multi-agent coordination. Specifically, we endow the student with population-invariant communication and a hierarchical skill set, allowing it to learn cooperation and behavior skills from distinct tasks with varying numbers of agents. In addition, we model the teacher as a contextual bandit conditioned by student policies, enabling a team of agents to change its size while still retaining previously acquired skills. We also analyze the inherent non-stationarity of this multi-agent automatic curriculum teaching problem and provide a corresponding regret bound. Empirical results show that our method improves the performance, scalability and sample efficiency in several MARL environments. The source code and the video can be found at https://sites.google.com/view/marl-spc/.

## 1 Introduction

Multi-agent reinforcement learning (MARL) has long been a go-to tool in complex robotic and strategic domains [1, 2]. However, learning effective policies with sparse reward from scratch for large-scale multi-agent systems remains challenging. One of the challenges is the exponential growth of the joint observation-action space with an increasing number of agents. In addition, sparse reward signal requires a large number of training trajectories, posing difficulties in applying existing MARL algorithms directly to complex environments. As a result, these algorithms may produce agents that do not collaborate with each other, even when it would be of significant benefit [3, 4].

There are several lines of research related to the large-scale MARL problem with sparse reward, including reward shaping [5], curriculum learning [6], and learning from demonstrations [7]. Among these approaches, the curriculum learning paradigm, in which the difficulty of experienced tasks and the population of training agents progressively grow, shows particular promise. In *automatic curriculum learning* (ACL), a teacher (curriculum generator) learns to adjust the complexity and sequencing of tasks faced by a student (curriculum learner). Several works have even proposed *multi-agent* ACL algorithms, based on approximate or heuristic approaches to teaching, such as DyMA-CL

[8], EPC [9], and VACL [6]. However, these approaches rely on a framework of an off-policy student with a replay buffer that is hard to decide the size of the replay buffer since the proportion of different tasks matters. Also, they make a strong assumption that the value of the learned policy does not change when agents switch to a different task. For example, In the football environment, when we treat the score as the reward, the same state-action pairs of the team agents in different tasks might lead to different returns. 3 learned agents could get more scores in a 3v1 match, while the same three agents could get fewer scores in a 4v11 match with an unlearned random teammate. When decomposing at the same state-action pairs, agents get different credit assignments. Moreover, the teacher in these approaches still faces a non-stationarity problem due to the ever-changing student strategies. Another class of larger-scale MARL solutions is hierarchical learning, which utilizes temporal abstraction to decompose a task into a hierarchy of subtasks. This includes skill discovery [10], option as response [11], role-based MARL [12], and two levels of abstraction [13]. However, these approaches mostly focus on one specific task with a fixed number of agents and do not consider the transferability of learned skills. In this paper, we provide our insight into this question:

*Whether an elaborate combination of principles from ACL and hierarchical learning can enable* **complex** *cooperation with* **sparse reward** *in* **MARL**?

Specifically, we present a novel automatic curriculum learning algorithm, Skilled Population Curriculum (SPC), that addresses the challenges of learning effective policies for large-scale multi-agent systems with sparse reward. The core idea behind SPC, motivated by real-world team sports where players often train their skills by gradually increasing the difficulty of tasks and the number of coordinating players, is to encourage the student to learn skills from tasks with different numbers of agents, akin to how team sports players train by gradually increasing the difficulty of tasks and the number of coordinating players. To achieve this, SPC is implemented with three key components. First, to solve the final complex cooperative tasks, we equip the contextual bandit teacher with an RNN-based [14] imitation model to represent student policies and generate the bandit's context. Second, to handle the varying number of agents across these tasks and bypass the limitation of the related studies, we utilize population-invariant communication in the student module is implemented to handle varying number of agents across tasks. By treating each agent's message as a word and using a self-attention communication channel [15], SPC supports an arbitrary number of agents to share messages. Third, to learn transferable skills in the sparse reward setting, a hierarchical skill framework is used in the student module to learn transferable skills in the sparse reward setting, where agents communicate on the high-level about a set of shared low-level policies. Empirical results show that our method achieves state-of-the-art performance in several tasks in Multi-agent Particle Environment (MPE) [16] and the challenging 5vs5 competition in Google Research Football (GRF) [17].

## 2  Preliminaries

**Dec-POMDP.** A cooperative MARL problem can be formulated as a *decentralized partially observable Markov decision process* (Dec-POMDP) [18], which is described as a tuple $\langle n, \boldsymbol{S}, \boldsymbol{A}, P, R, \boldsymbol{O}, \boldsymbol{\Omega}, \gamma \rangle$, where $n$ represents the number of agents. $\boldsymbol{S}$ represents the space of global states. $\boldsymbol{A} = \{A_i\}_{i=1,\cdots,n}$ denotes the space of actions of all agents. $\boldsymbol{O} = \{O_i\}_{i=1,\cdots,n}$ denotes the space of observations of all agents. $P : \boldsymbol{S} \times \boldsymbol{A} \to \boldsymbol{S}$ denotes the state transition probability function. All agents share the same reward as a function of the states and actions of the agents $R : \boldsymbol{S} \times \boldsymbol{A} \to \mathbb{R}$. Each agent $i$ receives a private observation $o_i \in O_i$ according to the observation function $\boldsymbol{\Omega}(s, i) : \boldsymbol{S} \to O_i$. $\gamma \in [0, 1]$ denotes the discount factor.

**Multi-armed Bandit.** Multi-armed bandits (MABs) are a simple but very powerful framework that repeatedly makes decisions under uncertainty. In this framework, a learner performs a sequence of actions and immediately observes the corresponding reward after each action. The goal is to maximize the total reward over a given set of $K$ actions and a specific time horizon $T$. The measure of success in MABs is often determined by the regret, which is the difference between the cumulative reward of an MAB algorithm and the best-arm benchmark. One well-known MAB algorithm is the Exp3 algorithm [19], which aims to increase the probability of selecting good arms and achieves a regret of $O(\sqrt{KT \log(K)})$ under a time-varying reward distribution. Another related concept is the contextual bandit problem [20], where the learner makes decisions based on prior information as the context.

## 3 Skilled Population Curriculum

In this section, we first provide a formal definition of the curriculum-enhanced Dec-POMDP framework, which formulates the MARL with curriculum problem under the Dec-POMDP framework. We then present our multi-agent ACL algorithm, Skilled Population Curriculum (SPC), as shown in Fig. 1. In the following subsections, we establish the curriculum learning framework in Sec. 3.1, and then present a contextual multi-armed bandit algorithm as the teacher to address the non-stationarity in Sec. 3.2. Lastly, we introduce the student with transferable skills and population-invariant communication to tackle the varying number of agents and the sparse reward problem in Sec. 3.3.

### 3.1 Problem Formulation

We consider environments from multi-agent automatic curriculum learning problems are equipped with parameterized task spaces and thus can be modeled as curriculum-enhanced Dec-POMDPs.

**Definition 3.1** (Curriculum-enhanced Dec-POMDP). A curriculum-enhanced Dec-POMDP is defined by a tuple $\langle \Phi, \mathcal{M} \rangle$, where $\Phi$ and $\mathcal{M}$ represent a task space and a Dec-POMDP, respectively. Given the task $\phi$, the Dec-POMDP $\mathcal{M}(\phi)$ is presented as $\left\{ n^\phi, \boldsymbol{S}^\phi, \boldsymbol{A}^\phi, P^\phi, r^\phi, O^\phi, \boldsymbol{\Omega}^\phi, \gamma^\phi \right\}$. The superscript $\phi$ denotes that the Dec-POMDP elements are determined by the task $\phi$. Note that task $\phi$ can be a few parameters of the environment or task IDs in a finite task space. *In a curriculum-enhanced Dec-POMDP, the objective is to improve the student's performance on the target tasks through the sequence of training tasks given by the teacher.*.

Let $\tau$ denote a trajectory whose unconditional distribution $\Pr_\mu^{\pi,\phi}(\tau)$ (under a policy $\pi$ and a task $\phi$ with initial state distribution $\mu(s_0)$) is $\Pr_\mu^{\pi,\phi}(\tau) = \mu(s_0) \sum_{t=0}^\infty \pi(a_t \mid s_t) P^\phi(s_{t+1} \mid s_t, a_t)$. We use $p(\phi)$ to represent the distribution of target tasks and $q(\phi)$ to represent the distribution of training tasks at each task sampling step. We consider the joint agents' policies $\pi_\theta(a|s)$ and $q_\psi(\phi)$ parameterized by $\theta$ and $\psi$, respectively. The overall objective to maximize in a curriculum-enhanced Dec-POMDP is:

$$J(\theta, \psi) = \mathbb{E}_{\phi \sim p(\phi), \tau \sim \Pr_\mu^\pi} \left[ R^\phi(\tau) \right] \quad = \mathbb{E}_{\phi \sim q_\psi(\phi)} \left[ \frac{p(\phi)}{q_\psi(\phi)} V(\phi, \pi_\theta) \right] \tag{1}$$

where $R^\phi(\tau) = \sum_t \gamma^t r^\phi(s_t, a_t; s_0)$ and $V(\phi, \pi_\theta)$ represents the value function of $\pi_\theta$ in Dec-POMDP $\mathcal{M}(\phi)$. However, when optimizing $q_\psi(\phi)$, we cannot get the partial derivative $\nabla_\psi J(\theta, \psi) = \nabla_\psi \sum_\tau \frac{1}{q_\psi(\phi)} R^\phi(\tau) \Pr_\mu^{\pi,\phi}(\tau)$[1] since the reward function and the transition probability function w.r.t number of agents are non-parametric, non-differentiable, and discontinuous in most MARL scenarios.

Thus, we use the non-differentiable method, i.e., multi-armed bandit algorithms, to optimize $q_\psi(\phi)$, and use an RL algorithm (the student) in alternating periods to optimize $\pi_\theta(a|s)$. However, there are three key challenges in solving this problem: (1) The teacher is facing a non-stationarity problem due to the ever-changing student's strategies. (2) The student will forget the old tasks and need to re-learn them. Some tasks can be the prerequisites of other tasks, while some can be inter-independent and parallel. (3) There is a lack of a general student framework to deal with the varying number of agents across tasks and the sparse reward problem.

### 3.2 Teacher as a Non-Stationary Contextual Bandit

As previously discussed, the teacher faces a non-stationarity problem due to the ever-changing student's strategies during the learning process. Specifically, as the student learns across different tasks in different learning stages, the teacher will observe varying student performance when providing the same task, resulting in a time-varying reward distribution for the teacher. In addition, the student may forget previously learned policies. To mitigate this problem, the teacher should balance the exploitation of tasks that have been found to benefit the student's performance on the target tasks, with the exploration of tasks that may not directly facilitate the student's learning.

Fortunately, we notice that the non-stationarity stems from the student, which can be mitigated with a contextual bandit which embeds the student policy into the context. As shown in Fig. 1 Left, the teacher utilizes the student's policy representation as the context and chooses a task from the

---

[1] $p(\phi)$ is not in the partial derivative since it is a fixed distribution.

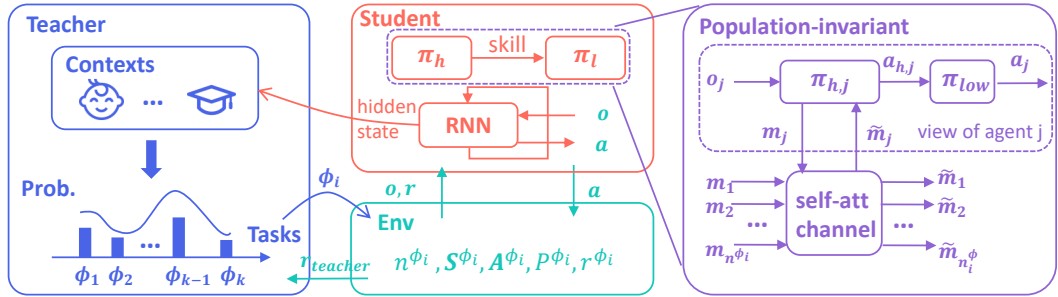

Figure 1: The overall framework of SPC. It consists of three parts: configurable environments, a teacher, and a student. **Left.** The teacher is modeled as a contextual multi-armed bandit. At each teacher timestep, the teacher chooses a training task from the distribution of bandit actions. **Mid.** The student is endowed with a hierarchical skill framework and population-invariant communication. It is trained with MARL algorithms on the training tasks. The student returns not only the hidden state of its RNN imitation model as contexts to the teacher, but also the average discounted cumulative rewards on the testing task. **Right.** The student learns hierarchical policies, with the population-invariant communication taking place at the high-level, implemented with a self-attention communication channel to handle the messages from a varying number of agents. The agents in the student share the same low-level policy.

distribution of training tasks. Specifically, we extend the Exp3 algorithm [19] by incorporating contexts through a two-step online clustering process [21]. The context, represented by $x$, is the student's policy representation. The teacher's action is a specific task, denoted by $\phi$, and the teacher's reward is the return of the student in the target tasks. The teacher's algorithm is outlined in Alg. 1. During the sampling stage (steps 1-5), the teacher selects a task for the student's training. In the training stage (steps 6-7), the teacher adjusts the parameters based on the evaluation reward received from the student.

---

**Algorithm 1** Teacher Sampling and Training

---

**Input:** Context $x$, the number of Clusters $N_c$, $N_c$ instances of Exp3 with task distribution $w(\phi_k, c)$ for $k = 1, \ldots, K$ and for $c = 1, \ldots, N_c$, learning rate $\alpha$, a buffer maintaining the historical contexts

**Output:** $\mathcal{M}(\phi) = \left\{ n^\phi, \boldsymbol{S}^\phi, \boldsymbol{A}^\phi, P^\phi, r^\phi, O^\phi, \boldsymbol{\Omega}^\phi, \gamma^\phi \right\}$, the teacher bandit parameters

**Sampling**
1. Get the the context $x$, and save it to the buffer
2. Run the online cluster algorithm and get the index of the cluster center $c(x)$
3. Let the active Exp3 instance be the instance with index $c(x)$
4. Set the probability $p(\phi_k, c(x)) = \frac{(1-\alpha)w(\phi_k, c(x))}{\sum_{j=1}^{K} w(\phi_k, c(x))} + \frac{\alpha}{K}$ for each task $\phi_k$
5. Sample a new task according to the distribution of $p_{\phi_k, c}$

**Training**
6. Get the return (discounted cumulative rewards) from student testing $r$
7. Update the active Exp3 instance by setting $w(\phi_k, c(x)) = w(\phi_k, c(x))e^{\alpha r/K}$

---

### 3.2.1 Context Representation

Upon analysis, it is essential to learn an effective representation for the student's policy as the context. One straightforward representation is to use the student parameters $\theta$ directly as the context. However, the number of parameters is too large to be used as the input of neural network if we change the student's architecture. Therefore, we propose an alternative method.

A principle for learning a good representation of a policy is *predictive representation*, which means the representation should be accurate to predict policy actions given states. In accordance with this principle, we utilize an imitation function through supervised learning. Supervised learning does not require direct access to reward signals, making it an attractive approach for reward-agnostic representation learning. Intuitively, the imitation function attempts to mimic low-level policy based

on historical behaviors. In practice, we use an RNN-based imitation function $f_{im} : \mathcal{S} \times \mathcal{A} \rightarrow [0, 1]$. Since recurrent neural networks are theoretically Turing complete [22], their internal states can be used as the representation of the student's policy. We train this imitation function by using the negative cross entropy objective $\mathbb{E}[\log f_{im}(s, a)]$.

### 3.2.2 Regret Analysis

In this subsection, we demonstrate that the proposed teacher algorithm has a regret bound of $\mathbb{E}[R(T)] = O\left(T^{2/3}(LK \log T)^{1/3}\right)$, where $T$ is the number of total rounds, $L$ is the Lipschitz constant, and $K$ is the number of arms (the number of the teacher's actions). The regret analysis is used to justify the usage of the bandit algorithm in the non-stationary setting. The regret bound represents the optimality of SPC, as the teacher's reward is the return of the student in the target tasks.

First, we introduce the Lipschitz assumption about the generalization ability of the task space.

**Assumption 3.2** (Lipschitz continuity w.r.t the context). Without loss of generality, the contexts are mapped into the $[0, 1]$ interval, so that the expected rewards for the teacher are Lipschitz with respect to the context.

$$|r(\phi \mid x) - r(\phi \mid x')| \leq L \cdot |x - x'|$$
for any arm $\phi \in \Phi$ and any pair of contexts $x, x' \in \mathcal{X}$
$$(2)$$

where $L$ is the Lipschitz constant, and $\mathcal{X}$ is the context space.

This assumption suggests that for any policy trained on a set of tasks, the rate at which performance improves is not faster than the rate at which the policy changes. This is a realistic assumption, as we cannot expect the student to achieve a significant improvement on a task with only a few training steps under a new context. We use an existing contextual bandit algorithm for a limited number of contexts [19] (see Appendix A) and Lemma 3.3 as a foundation for proving Theorem 3.4.

**Lemma 3.3.** *Alg. 2 has a regret bound of* $\mathbb{E}[R(T)] = \mathcal{O}(\sqrt{TK|\mathcal{X}| \log K})$.

Lemma 3.3 introduces a square root dependence on $|\mathcal{X}|$ if separate copies of Exp3 are run for each context [19]. This motivates us to address the large context space by utilizing discretization techniques.

**Theorem 3.4.** *Consider the Lipschitz contextual bandit problem with contexts in* $[0, 1]$. *The Alg. 1 yields regret* $\mathbb{E}[R(T)] = O\left(T^{2/3}(LK \ln T)^{1/3}\right)$.

*Proof.* See Appendix B. $\qquad\square$

In practice, the high-dimensional context space cannot be discretized using a uniform mesh in $[0, 1]$ as in the proof of Theorem 3.4. To address this issue, we utilize the Balanced Iterative Reducing and Clustering using Hierarchies (BIRCH) online clustering algorithm [21] to discretize the context space. BIRCH is an efficient and easy-to-update algorithm that can effectively cluster large datasets. In this case, it is used to cluster the high-dimensional RNN-based policy representation. The resulting clusters can be seen as an approximation of a uniform mesh.

### 3.3 Student with Population-Invariant Skills

We propose a population-invariant skill framework to address the challenges of varying number of agents and sparse reward problem. This framework allows agents to communicate via a self-attention channel, enabling them to learn transferable skills across different tasks. The student module is designed to be algorithm-agnostic and is orthogonal to any state-of-the-art MARL algorithm. While there have been some efforts in the literature to address the varying number of agents [23, 24], these approaches heavily rely on prior knowledge of the environments.

**Population-Invariant Teamwork Communication.** In order to enable the population-invariant property and learn tactics among agents, we introduce communication. Leveraging the transformer architecture's capability to process inputs of varying lengths [15], we incorporate self-attention into our communication mechanism. As illustrated in Fig. 1 Right, each agent $j$ receives an observation $o_j$ and encodes it into a message vector $m_j = f(o_j)$ which is then sent through a self-attention channel, where $f$ is an observation encoder function.

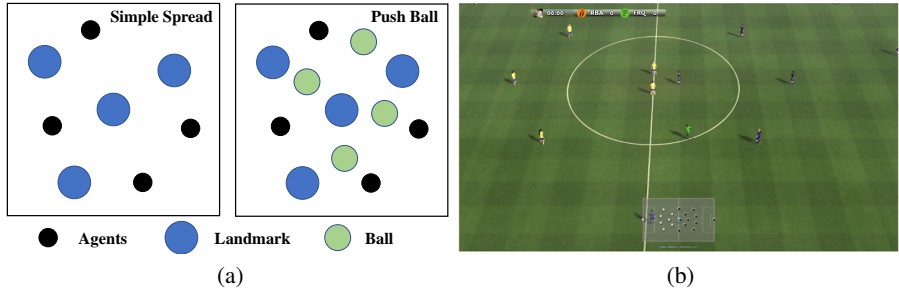

Figure 2: (a) Multi-agent Particle Environment. (b) Google Research Football.

The channel aggregates all messages and sends the new message vector, $\tilde{m}_j$, through the self-attention mechanism. Concretely, given the channel input $\mathbf{M} = [m_1, m_2, \cdots, m_n] \in R^{n \times d_m}$, and the trainable weight of the channel $\mathbf{W}_Q, \mathbf{W}_K, \mathbf{W}_V \in R^{d_m \times d_m}$, we obtain three distinct representations: $\mathbf{Q} = \mathbf{M}\mathbf{W}_Q, \mathbf{K} = \mathbf{M}\mathbf{W}_K, \mathbf{V} = \mathbf{M}\mathbf{W}_V$. Then the output messages are

$$\tilde{\mathbf{M}} = \text{Attention}(\mathbf{Q}, \mathbf{K}, \mathbf{V}) = \text{softmax}\left(\frac{\mathbf{Q}\mathbf{K}^T}{\sqrt{d_m}}\right)\mathbf{V} \tag{3}$$

where $d_m$ is the dimension of the messages. As the dimensions of the trainable weight are independent of the number of agents, our student models can leverage the population-invariant property to effectively learn tactics.

**Transferable Hierarchical Skills.** As depicted in the dotted box in Fig. 1 Right, after receiving the new messages $\tilde{m}_j$ from the channel, each agent employs a high-level action (skill) $a_{h,j} = \pi_{h,j}(o_j, \tilde{m}_j)$ to execute the low-level policy $a_j = \pi_{low}(o_j, a_{h,j})$. In this work, we generalize the high-level action (skill) $a_{h,j}$ to a continuous embedding space, so that the skill can be either a latent continuous vector as in DIAYN [25], or a categorical distribution for sampling discrete options [26].

**Implementation.** We implement the high- and low-level policies in the student with Proximal Policy Optimization (PPO) [27]. Following the common practice proposed in [28], the high-level policy for each agent is learned independently, whereas the low-level policies share parameters, as the fundamental action pattern should be consistent among different agents. The low-level agents are rewarded by the environment, while the high-level policy is trained to take actions at fixed intervals. Within this interval, the cumulative low-level reward is used as the high-level reward. When using a categorical distribution to enable discrete skills, we sample an "option" from the distribution and provide the corresponding one-hot embedding to the low-level policy.

## 4 Related Work

**Automatic Curriculum Learning in MARL.** Curriculum learning is a training strategy that mimics the human learning process by organizing tasks based on their difficulty level [29]. The selection of tasks is formulated as a Curriculum Markov Decision Process (CMDP) [30]. Automatic Curriculum Learning mechanisms aim to learn a task selection function based on past interactions, such as ADR [31, 32], ALP-GMM [33], SPCL [34], GoalGAN [35], PLR [36, 37], SPDL [38], CURROT [39], and graph-curriculum [40]. Recently, several MARL curriculum learning frameworks have been proposed, such as open-ended evolution [41–43], population-based training [44, 45], meta-learning [46, 47] and training with emergent curriculum [48, 49, 29]. In summary, these frameworks share a common principle of an automatic curriculum that continually generates improved agents through selection pressure among a population of self-optimizing agents.

**Hierarchical MARL and Communication.** Hierarchical reinforcement learning (HRL) has been extensively studied to address the issue of sparse reward and facilitate transfer learning. Single-agent HRL focuses on learning the temporal decomposition of tasks, either by learning subgoals [50–54] or by discovering reusable skills [55–58]. Recent developments in hierarchical MARL have been discussed in Sec. 1. In multi-agent settings, communication has been effective in promoting cooperation among agents [59–65]. However, current approaches that extend HRL to multi-agent systems or utilize communication are limited to a fixed number of agents and lack the ability to transfer to different agent counts.

## 5 Experiments

To demonstrate the effectiveness of our approach, we conduct experiments on several tasks in two environments: Simple-Spread and Push-Ball in the Multi-agent Particle Environment (MPE) [16], and the challenging 5vs5 task of the Google Research Football (GRF) environment [17]. We aim to investigate the following research questions:

**Q1**: *Is curriculum learning necessary in complex large-scale MARL problems?* (Sec. 5.2)

**Q2**: *Can SPC outperform previous curriculum-based MARL methods? If so, which components of SPC contribute the most to performance gains?* (Sec. 5.3)

**Q3**: *Can SPC effectively learn a curriculum for the student?* (Sec. 5.4)

### 5.1 Environments, Baselines and Metric

**Environments.** In the GRF 5vs5 scenario, we control four agents, excluding the goalkeeper, to compete against the built-in AI opponents. Each agent observes a compact encoding, consisting of a 115-dimensional vector that summarizes various aspects of the game, such as player coordinates, ball possession and direction, active players, and game mode. The available action set for an individual agent includes 19 discrete actions, such as idle, move, pass, shoot, dribble, etc. The GRF provides two types of rewards: scoring and checkpoints, to encourage agents to move the ball forward and make successful shots. Additionally, we include a shooting reward in the challenging GRF 5vs5 task. We select several basic scenarios in GRF, including 3vs3, Pass-Shoot, 3vs1, and Empty-Goal as curriculum.

In MPE, we investigate Simple-Spread and Push-Ball (see Fig. 2a). In Simple-Spread, there are $n$ agents that need to cover all $n$ landmarks. Agents are penalized for collisions and only receive a positive reward when all the landmarks are covered. In Push-Ball, there are $n$ agents, $n$ balls, and $n$ landmarks. The agents must push the balls to cover each landmark. A success reward is given after all the landmarks have been covered.

**Baselines.** We compare our approach to the following methods in Table 1 as baselines[2]:

**Metric.** To evaluate the performance of our approach in the GRF 5vs5 scenario, we use metrics beyond just the mean episode reward, as this alone may not accurately reflect the agents' performance. Specifically, we use the win rate and the average goal difference, which is calculated as the number of goals scored by the MARL agents minus the number of goals scored by the opposing team.

Table 1: Baseline algorithms.

| Categories | Methods |
|---|---|
| MARL (**Q1**) | QMIX [68] IPPO [69] |
| Curriculum-based (**Q2**) | IPPO with uniform task sampling VACL [6] |
| Ablation Study (**Q3**) | SPC with uniform task sampling SPC without HRL and COM |

We evaluate the performance of MARL algorithms to justify the need for curriculum learning in complex large-scale MARL problems. To ensure a fair comparison, we modify VACL by removing the centralized critic for MPE tasks. Centralized Training Decentralized Execution methods is not included as baselines since they are not suitable for varying numbers (e.g., MADDPG/MAPPO's critic requires a fixed size of input or QMIX's mixing network also fixed size of the input).

In all experiments, we use individual Proximal Policy Optimization (IPPO) as the backend MARL algorithm. To ensure the robustness of our results, we conduct experiments on a 30-node cluster, with one node containing a 128-core CPU and four A100 GPUs. Each trial of the experiment is repeated over five seeds and runs for 1-2 days.

### 5.2 The Necessity of Curriculum Learning

Our experiments first show that in simple environments, such as MPE, students can directly learn to complete the task without the need for curriculum. For MPE experiments, we randomly select a starting state and the episode ends after a fixed number of maximum steps. Specifically, the task

---

[2]We also run CDS [66] and CMARL [67], but we have not included their performance because the goal difference reported in CMARL [67] is relatively low compared to our method.

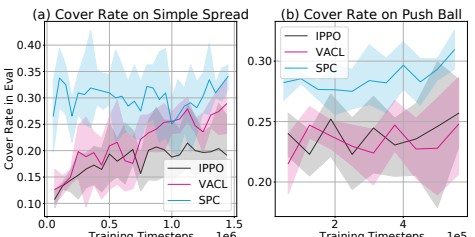
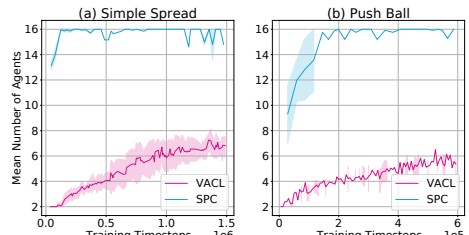

Figure 3: The evaluation performance of various methods on MPE.

Figure 4: The changes in the number of agents on MPE.

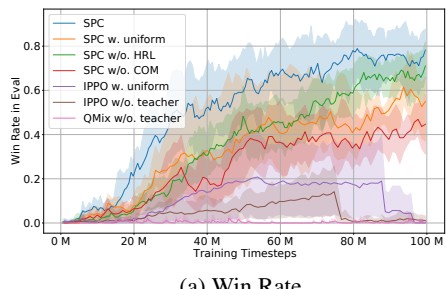
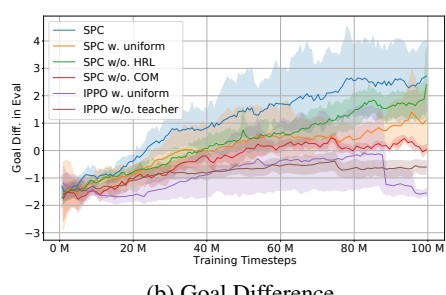

(a) Win Rate

(b) Goal Difference

Figure 5: The evaluation performance of various methods on 5vs5 football competition. (p-value is less than 0.05 which means the results are statistically significant.)

space consists of $n$ agents, where $n \in \{2, 4, 8, 16\}$, and the maximum allowed steps is set to 25. All evaluations are performed on the target task, with $n = 16$. IPPO is trained and evaluated directly on the target task, and results in Fig. 3 demonstrate that it performs similarly to the VACL algorithm. We plot the performance within a sliding window so that the starting point is not exactly from 0 timestep. VACL uses entity progression, which is a rule-based curriculum update mechanism so it lacks the flexibility to switch the curriculum when relatively easy tasks can be learned quickly. The reason for the performance jump is that SPC can switch to the largest population rapidly, which we consider one advantage of SPC. Additionally, we observe that the SPC approach only achieves a slightly higher coverage rate than the baseline methods. Furthermore, we investigate the probability variation of different population sizes, shown in Fig. 4. We observe that the curriculum provided by SPC is approaching the target task. These results suggest that in simple environments where the student can learn to directly complete the task, curriculum learning may not be necessary.

When it comes to more complex scenarios, such as the 5vs5 task in GRF, our results demonstrate that curriculum learning is a promising solution. As shown in Fig. 5a, without curriculum learning, QMix and IPPO cannot perform well in the 5vs5 scenario, and IPPO is slightly better than QMix. In Fig. 5b, we omit the curve of QMix as its mean score is low and affects the presentation of the figure. The reason could be that QMix is an off-policy MARL algorithm, which would rely heavily on the replay buffer. However, in such sparse reward scenarios, the replay buffer has much less effective samples for QMix to learn. For example, the replay buffer would contain tons of zero-score samples, leading to a non-promising performance. Meanwhile, IPPO, with its on-policy nature, is able to achieve better sample efficiency and outperform off-policy algorithms like QMix in such scenarios. Though MARL methods can achieve good performance in basic scenarios in GRF, they fail to solve complex scenarios such as the 5vs5 task. Therefore, curriculum learning is a promising solution to the complex large-scale MARL problem.

### 5.3 Performance and Ablation Study

Our study demonstrates that SPC outperforms VACL in MPE tasks. Instead of training with a continuous relaxation of the population size variable as in VACL, our bandit teacher achieves a higher success rate at test time, since the population size is a discrete variable in nature. Furthermore, the curriculum provided by SPC is effective in exploring the task space and converge to the target task when the task is relatively simple and curriculum is not necessary, as shown in Fig. 4.

In GRF experiments, we do not include VACL in our baselines in the GRF, as its implementation relies heavily on prior knowledge of specific scenarios, such as the thresholds to divide the learning

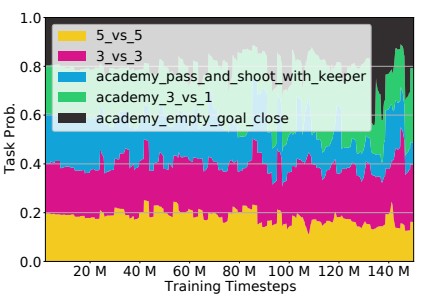 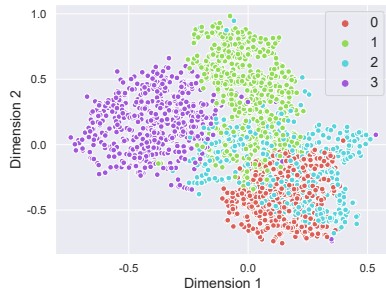

(a) The task distribution of SPC during training.    (b) The visualization of contexts

Figure 6: Visualization of Learned Curriculum.

process. Fig. 6 indicates that SPC has higher win rate and goal difference than IPPO with uniform task sampling in the 5vs5 competition. These experiments demonstrate that when the teacher is rewarded by the student's performance, a bandit-based teacher can exploit the student's learning stage and provide suitable training tasks.

In our ablation study, we examine the impact of two key components of our SPC algorithm: the contextual multi-armed bandit teacher and the hierarchical structure of the student framework. By replacing the former with uniform task sampling and removing the latter, As shown in Fig. 5a and Fig. 5b, SPC can achieve a higher win rate and a greater score difference than SPC with uniform and SPC without HRL. Furthermore, SPC with uniform task sampling outperforms IPPO with uniform task sampling. This highlights the importance of HRL in the 5vs5 football competition, and suggests that both the contextual multi-armed bandit and the hierarchical structure contribute equally to the performance of SPC. When removing HRL and bandit, the performance degradation w.r.t. SPC are similar. However, it should be noted that SPC with uniform task sampling has a larger variance in performance than SPC without HRL, indicating that uniform sampling may introduce more undesired tasks for student training. Overall, these results further justify the necessity of SPC in complex large-scale MARL problems[3].

## 5.4 Visualization of Learned Curriculum

We visualize the distribution of task sampling of SPC during training based on a selected trial as shown in Fig. 6a. At the beginning of training, the task probability appears to be near-uniform, as the teacher explores the task space and keeps track of the student's learning status, acting as an anti-forgetting mechanism. As training progresses, the probabilities change over time. For example, the proportions of 3vs1 and Empty-Goal tasks gradually drop as the student becomes proficient in these scenarios. We also visualize the distribution of contexts in Fig. 6b using t-SNE [70], where the contexts are collected and stored in a buffer. We divide the contexts into four classes according to the index, and different parts represent different contexts of the final student policy representation.

## 6 Discussion

**Conclusion.** We present Skilled Population Curriculum (SPC), a novel multi-agent ACL algorithm that addresses scalability and sparse reward issues in multi-agent systems. SPC learns complex behaviors from scratch by incorporating a population-invariant multi-agent communication framework and using a hierarchical scheme for agents to learn skills. Moreover, SPC mitigates non-stationarity by modeling the teacher as a contextual bandit, where the context is represented by the student's policy representation. Though our design choices focus on solving the GRF 5vs5 task, we believe that analyzing and addressing these issues is crucial for further development in multi-agent ACL algorithms. While SPC may be complex to implement due to its various components, we provide clean and well-organized code for ease of use.

**Limitations.** We acknowledge that there are limitations of our algorithm. SPC is over-designed for simple tasks since our objective is to solve difficult tasks. Also, it would be interesting to understand the impact of varying number of agents on the dynamics of the environment.

---

[3]We also demonstrate the performance of SPC in the GRF 11vs11 full game (see Appendix C).

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

## A Contextual Bandit for Limited Number of Contexts

---

**Algorithm 2** A contextual bandit algorithm for a small number of contexts

---
1: **Initialization:** For each context $x$, create an instance $\text{Exp3}_x$ of algorithm Exp3
2: **for** round **do**
3:     Invoke algorithm $\text{Exp3}_x$ with $x = x_t$
4:     Play the action chosen by $\text{Exp3}_x$
5:     Return reward $r_t$ to $\text{Exp3}_x$
6: **end for**

---

## B Proof of Theorem 3.4

**Theorem 3.4.** *Consider the Lipschitz contextual bandit problem with contexts in $[0,1]$. The Alg. 1 yields regret $\mathbb{E}[R(T)] = O\left(T^{2/3}(LK \ln T)^{1/3}\right)$.*

*Proof.* Let $S_m$ be the $\epsilon$-uniform mesh on $[0,1]$, that is, the set of all points in $[0,1]$ that are integer multiples of $\epsilon$. We take $\epsilon = 1/(d-1)$ where the integer $d$ is the number of points in $S_m$, which will be adjusted later in the analysis.

We apply Alg. 2 to the context space $S_m$. Let $f_{S_m}(x)$ be a mapping from context $x$ to the closest point in $S_m$:

$$f_{S_m}(x) = \min\left(\underset{x' \in S_m}{\operatorname{argmin}} |x - x'|\right)$$

In each round $t$, we replace the context $x_t$ with $f_{S_m}(x_t)$ and call $\text{Exp3}_S$. The regret bound will have two components: the regret bound for $\text{Exp3}_S$ and (a suitable notion of) the discretization error. Formally, let us define the "discretized best response" $\pi^*_{S_m} : \mathcal{X} \to \Phi$: $\pi^*_{S_m}(x) = \pi^*\left(f_{S_m}(x)\right)$ for each context $x \in \mathcal{X}$.

We define the total reward of an algorithm Alg is $\text{Reward}(\text{Alg}) = \sum_{t=1}^{T} r_t$. Then the regret of $\text{Exp3}_S$ and the discretization error are defined as:

$$R_S(T) = \text{Reward}(\pi^*_S) - \text{Reward}(\text{Exp3}_S)$$
$$\text{DE}(S) = \text{Reward}(\pi^*) - \text{Reward}(\pi^*_S).$$

It follows that regret is the sum $R(T) = R_S(T) + \text{DE}(S)$. We have $\mathbb{E}[R_S(T)] = \mathcal{O}(\sqrt{TK \log K})$ from Lemma 3.3, so it remains to upper bound the discretization error and adjust the discretization step $\epsilon$.

For each round $t$ and the respective context $x = x_t$, $r\left(\pi^*_S(x) \mid f_S(x)\right) \geq r\left(\pi^*(x) \mid f_S(x)\right) \geq r\left(\pi^*(x) \mid x\right) - \epsilon L$. The first inequality is determined by the optimality of $\pi^*_S$ and the second is determined by Lipschitzness. Summing this up over all rounds $t$, we obtain $\mathbb{E}[\text{Reward}(\pi^*_S)] \geq \text{Reward}[\pi^*] - \epsilon LT$.

Thus, the regret is that

$$\mathbb{E}[R(T)] \leq \epsilon LT + O\left(\sqrt{\frac{1}{\epsilon}TK \log T}\right) = O\left(T^{2/3}(LK \log T)^{1/3}\right) \tag{4}$$

For the last inequality, we want the two terms of the regret bound has the same asymptotic complexity. So when $\epsilon LT = \operatorname{sqrt}\frac{1}{\epsilon}TK \log T$, we can get $\epsilon = \left(\frac{K \log T}{TL^2}\right)^{1/3}$. So, we choose $\epsilon = \left(\frac{K \log T}{TL^2}\right)^{1/3}$. $\square$

## C SPC on GRF 11vs11 Full Game

We also conduct experiments on the GRF 11vs11 full game scenario with sparse reward. As shown in Fig. 7, SPC achieves about 50% win rate against built-in AI in the target task after training with 200

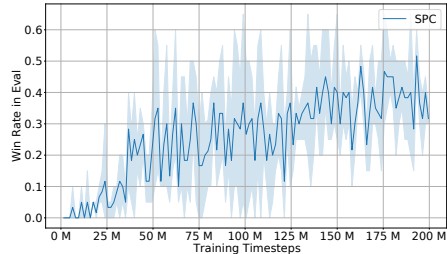

Figure 7: The performance of SPC on the 11v11 scenario.

million timesteps. This is non-trivial as this is one of the most challenging benchmarks for MARL community, and most current MARL methods struggle to achieve progress without hand-crafted engineering.

## D Qualitatively Analysis On Low-Level Skills

We demonstrate game statistics under different high-level actions. For example, the times of shooting, passing and running actions per game in GRF. These different low-level policies are induced by the high-level actions. We evaluate these statistics by fixing one agent's high-level actions and maintaining other agents with SPC. The results in Table 2 are averaged over five runs in the 5vs5 scenario.

Table 2: Statistics of low-level skills.

|         | shooting per game | passing per game | running per game   |
|---------|-------------------|------------------|--------------------|
| skill 1 | 7.9 times         | 0.5 times        | 2254 time steps    |
| skill 2 | 2.3 times         | 26.4 times       | 2149 time steps    |
| skill 3 | 1.6 times         | 3.9 times        | 2875 time steps    |

## E Comparing Different Teacher Algorithms on GRF Corner-5

To further illustrate the effectiveness of the SPC teacher module, we conduct experiments on the corner-5 scenario on GRF, where the target task is to control five of the eleven players to obtain a goal in the GRF Corner scenario. The experiments are designed to determine whether or not the contextual bandit in SPC outperforms alternative curriculum learning methods to schedule the number of agents in training. We compare SPC teacher against non-curriculum training (None), uniform task sampling (Uniform), a state-of-the-art curriculum learning method (ALP-GMM), and a

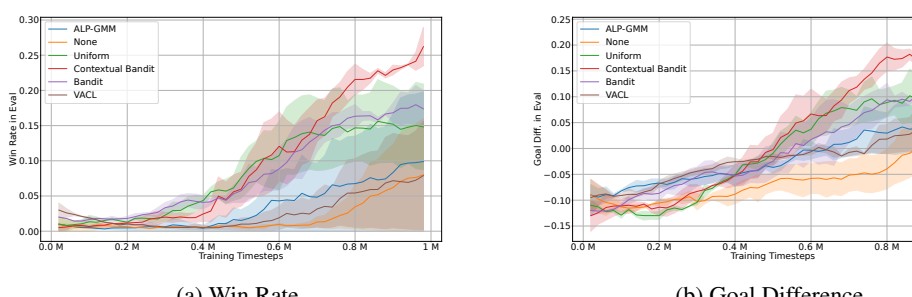

(a) Win Rate      (b) Goal Difference

Figure 8: The evaluation performance of various teacher algorithms on the GRF corner-5 scenario.

multi-agent curriculum learning method (VACL). The training task space consists of $n$ agents, where $n \in \{1, 3, 5\}$. All teachers have the same base architecture without transformer architecture and HRL. We also investigate the ablation of the RNN-based contexts (see Contextual Bandit and Bandit). Fig. 8 shows the benefit of SPC contextual bandit over other ACL methods after training with one million timesteps.

# F  Implementation Details

We use the default implementation of Proximal Policy Optimization (PPO) in Ray RLlib, which scales out using multiple workers for experience collection. This allows us to use a large amount of rollouts from parallel workers during training to ameliorate high variance and aid exploration. We do multiple rollouts in parallel with distributed workers and use parameter sharing for each agent. The trainer broadcasts new weights to the workers after their synchronous sampling.

## F.1  Google Research Football

We set five tasks for training the GRF 5vs5 scenario, including 5vs5, 3vs3, Pass-Shoot, 3vs1, and Empty-Goal. In the Empty-Goal, one agent need to move forward and shoot with an empty goal. In Pass-Shoot and 3vs3, two agents are controlled to play against a goalkeeper and three players, with different position initialization. In 3vs1, three agents are controlled to play against a center-back and a goalkeeper. In 5vs5, four agents are controlled to play against five players. Without loss of generality, we initialize all player with fixed positions and roles as center midfielders.

We use both MLP and self-attention mechanism for the high-level policy, and use MLP for the low-level policy. For high-level policy, the input is first projected to an embedding using two hidden layers with 256 units each and ReLU activation, which is then fed into multi-head self-attention (8 heads, 64 units each). The output is then projected to the actions and values using another fully connected layer with 256 units. For low-level policy, we use MLP with two hidden layers with 256 units each, i.e., the default configuration of policy network in RLlib.

Table 3: SPC hyper-parameters.

(a) SPC hyper-parameters used in GRF.

| Name | Value |
| --- | --- |
| Discount rate | 0.99 |
| GAE parameter | 1.0 |
| KL coefficient | 0.2 |
| Rollout fragment length | 1000 |
| Training batch size | 100000 |
| SGD minibatch size | 10000 |
| # of SGD iterations | 60 |
| Learning rate | 1e-4 |
| Entropy coefficient | 0.0 |
| Clip parameter | 0.3 |
| Value function clip parameter | 10.0 |

(b) SPC hyper-parameters used in MPE.

| Name | Value |
| --- | --- |
| Discount rate | 0.99 |
| GAE parameter | 1.0 |
| KL coefficient | 0.5 |
| # of SGD iterations | 10 |
| Learning rate | 1e-4 |
| Entropy coefficient | 0.0 |
| Clip parameter | 0.3 |
| Value function clip parameter | 10.0 |

## F.2  MPE

In MPE tasks, agents must cooperate through physical actions to reach a set of landmarks. Agents observe the relative positions of other agents and landmarks, and are collectively rewarded based on the proximity of any agent to each landmark. In other words, the agents have to cover all of the landmarks. Further, the agents are penalized when colliding with each other. The agents need to infer the landmark to cover and move there while avoid colliding with other agents.

The hyper-parameters of SPC in MPE are shown in Table 3b. In MPE, hyper-parameters such as rollout fragment length, training batch size and SGD minibatch size are adjusted according to horizon

615 of the scenarios so that policy are updated after episodes are done. We use the same neural network
616 architecture as in GRF, but with 128 units for all MLP hidden layers. Other omitted hyper-parameters
617 follow the default configuration in RLlib PPO implementation.

