# OpenReview forum: "Towards Skilled Population Curriculum for Multi-Agent Reinforcement Learning"
_NeurIPS.cc/2023/Conference — Submitted to NeurIPS 2023_

### Official Review · Reviewer_iWBE · 2023-07-05

**Soundness:** 3 good
**Presentation:** 2 fair
**Contribution:** 2 fair
**Rating:** 5
**Confidence:** 2

**Summary:**

This paper presents a new approach to automatic curriculum learning designed specifically for multi-agent coordination problems.

**Strengths:**

- The main strength of the paper, in my opinion, is the well-formulated approach to the curriculum learning problem. To the best of my knowledge of the related literature, the non-stationary contextual bandit as the teacher and the population-invariant skills for the students are both original and useful contributions to the literature.

**Weaknesses:**

- My general problem with this paper is that I am finding it hard to evaluate the significance of the work in the automatic curriculum learning sphere without an adequate baseline provided for GRF. Whilst the authors do argue that VACL is not used on GRF due to requiring prior knowledge, it seems unreasonable to therefore provide no baselines that are actually designed for these larger settings. For example, if VACL was unusable, then I would have maybe liked to have seen a comparison to population-based approaches in MARL or any of the other automatic curriculum learning approaches mentioned in Sec. 4. Overall, it is hard to properly evaluate the gains from this automatic curriculum learning framework without seeing the performance of baselines in an environment that actually requires automatic curriculum learning (MPE does not need it according to line 297-298).

I am happy to update my score if the authors can make a reasonable argument against the lack of other baselines in the work.

**Questions:**

- Line 37-39, 'However,..., tasks matters' - I am very confused by this sentence, could the authors please clarify?

- Upon inspection of the results, the final improved performance does not seem to be massively impacted by the hierarchical RL element of the framework. I was wondering if the authors could discuss a little more on this, in terms of its necessity in the framework and when they believe it would provide greater gains in performance?

**Limitations:**

The authors briefly make mention to the limitations of the work. I agree with the over-design of the framework for simple tasks, so would definitely like to see its performance in more difficult environments that it is designed for.

---

> ### Author Rebuttal · Authors · 2023-08-09
>
> Dear Reviewer iWBE,
>
> Thanks for your review of our paper. Here's our response to the weaknesses, questions, and limitations you have highlighted:
>
> -----
>
> **Q:** Make a reasonable argument against the lack of other baselines in the work
>
> **A:** Thanks for pointing this out. We compared different teacher/auto-curricula algorithms on GRF Corner-5 in Appendix D. The training task space consists of n agents, where n ∈ {1, 3, 5}. All teachers have the same base architecture without transformer architecture and HRL.
>
> -----
>
> **Q:** Clarification on 'However, these approaches (DyMA-CL, EPC, and VACL) rely on a framework of an off-policy student with a replay buffer that is hard to decide the size of the replay buffer since the proportion of different tasks matters.'
>
> **A:** It is hard to decide the size of the replay buffer since the proportion of different tasks matters. To be more specific, in the football environment, when we treat the score as the reward, the same state-action pairs of the team agents in different tasks might lead to different returns. For example, three learned agents could get more scores in a 3v1 match, while the same three agents could get fewer scores in a 4v11 match with an unlearned random teammate. When decomposing at the same state-action pairs, agents will get different credit assignment.
>
> We choose IPPO, a simple but efficient on-policy RL algorithm, as SPC’s backbone. In the same case, each agent will get its own reward, so that SPC doesn’t utilize a replay buffer and require the same assumption.
>
> -----
>
> **Q:** Upon inspection of the results, the final improved performance does not seem to be massively impacted by the hierarchical RL element of the framework. I was wondering if the authors could discuss a little more on this, in terms of its necessity in the framework and when they believe it would provide greater gains in performance?
>
> **A:** The hierarchical RL is used to extract useful skills for the student to transfer between the tasks. Currently, we test on GRF 5v5, and the learned skills are relatively simple, e.g., in Appendix D, shooting, passing, and running. We expected HRL will contribute more when the number of agents increases and more high-level strategies/tactics are emergent.
>
> -----
>
> We thank you for your endorsement of the motivation and the experimental studies of our work We appreciate your positive comments on the quality and experimental studies of our work. If you have any further questions or comments, we will be happy to discuss or fix them further. We are looking forward to your feedback.

---

> > ### Comment · Reviewer_iWBE · 2023-08-19
> >
> > Thank you to the authors for addressing my issue with the baselines by pointing out some results to me. I have updated my score to reflect this.

---

### Official Review · Reviewer_k74L · 2023-07-06

**Soundness:** 3 good
**Presentation:** 4 excellent
**Contribution:** 3 good
**Rating:** 6
**Confidence:** 4

**Summary:**

This paper presents Skilled Population Curriculum (SPC) which is a method for learning a curriculum to help a team of agents complete a complex task. SPC models the problem of choosing tasks for agents as a contextual bandit problem, and builds on top of the Exp3 algorithm to solve this bandit problem. SPC also uses an attention-based communication approach, and a hierarchical policy framework. Experiments are performed in the Multi-agent Particle Environment (MPE) and Google Research Football environment (GRF). While MPE does not seem to benefit much from SPC, in the more complex GRF domain the authors show using their SPC approach can accelerate training relative to MARL baselines.

**Strengths:**

- The paper is clear: it is well-structured, and provides a good balance of intuition and detail. It is clearly motivated, and addresses an interesting problem.

- The presented results show clear benefits to the authors' approach.

- The authors use suitable baselines approaches, and suitable environments.

**Weaknesses:**

- SPC seems to add significant computational complexity vs. baselines like IPPO. While the authors can justify focusing on sample complexity, for completeness they should also record information about the wall-clock time / computational resources needed to train their different baselines.

- In their research question stated on lines 52–53, the authors highlight their desire to consider complex sparse-reward settings. However, the MPE domains are a sparse setting, though fairly simple, and the results here show little benefit to using SPC. On the other hand, the GRF experiments appear to have a somewhat dense reward (the GRF checkpoint reward, which while not necessarily active every timestep, it could be argued is 'somewhat dense'). It seems like absent the checkpoint reward, SPC would struggle because there would be little information in the returns for the teacher agent to use — and I expect this would be the case in most complex (very) sparse reward environments

- Line 277 the authors state MADDPG/MAPPO would not be suitable in these experiments. This might be true in general, but for GRF specifically the critic input size actually would be the same across all tasks (as it pads observations if agents are absent). But since GRF is fully observable, MAPPO is equivalent to IPPO so this is not an issue for this work — though the authors may wish to revise their statement.

- Though the GRF environment is complex and difficult to solve, its level of complexity is somewhat deceptive, as evidenced by the video on the project website. The rollouts show that the agents have learned a simple "force an offside and run in a straight at the goal line" strategy which exploits deficiencies in the GRF bots. This behaviour has been observed before by Song et. al (http://arxiv.org/abs/2305.09458). However, this is not a fault of the authors, and is more broadly an issue in the MARL research community. Because of this, it's not clear what skills the agents learn in the training tasks that are useful in the target task. It would be interesting to see a plot of training task performance throughout training.

- It's unclear why the IPPO baselines have a sharp step change in performance around 80 and 90 million timesteps. The authors should investigate this, and perhaps make a comment (at least in the appendix) about why it occurs. In my experience, things like this sometimes occur when training runs stop unexpectedly before the full 100M timesteps, and so the remaining timesteps are aggregating over fewer seeds with lower performance. I would encourage the authors to produce plots reporting the interquartile mean of their results, and produce a plot showing the disaggregated training curves for each seed. These can go in the appendix.

- The authors state (line 301): " InFig. 5b, we omit the curve of QMix as its mean score is low and affects the presentation of the figure". I don't expect QMix to perform worse than the presumably near-uniform policies at the start of training for the other agents. So it's not clear how including QMix would disrupt the graph. Is it the case that QMix has a worse average goal difference than -2?

- Can the authors clarify: the target distribution for GRF is "100% 5vs5"? What is the target distribution for the MPE tasks? (I see now that these are mentioned later in the text: they should be mentioned when introducing the environments)

- It doesn't seem like there's a pattern to the task distribution (Fig. 6a) beyond "academy_pass_and_shoot_with_keeper becomes less common". It would be good to see the same plot for other trials. This possibly explainable by academy_pass_and_shoot_with_keeper requiring coordinated passing and shooting, whereas the 5vs5 rollouts (see video on project website) show a very simple GRF-bot exploiting strategy which does not closely resemble the behaviour required in academy_pass_and_shoot_with_keeper.

	- Where the authors claim "For example, the proportions of 3vs1 and Empty-Goal tasks gradually drop as the student becomes proficient in these scenarios", it is difficult to support this by looking at Fig. 6a.

- In my opinion this approach is over-engineered, but the authors do acknowledge this. Stripping some components (e.g the hierarchical RL) and focusing on deeply investigating the remaining components would improve this work

- Minor writing fixes:

	- line 42/43 "more scores" → "more goals"

	- line 43: "4v11" → "4v1" (I assume)

	- line 305: "tons of" → "many" (more formal tone)

**Questions:**

- Why did you decide to add the shooting reward for 5vs5? Does it make a big difference?

- What causes the step-change drop in IPPO performance?

- How does including QMix in 5b disrupt the graph?

**Limitations:**

- The limitations section is quite limited, and limitations and assumptions could be more clearly stated throughout.

- The authors recognise that their approach is complex and computationally intensive, so might not be applicable in simple environments. Testing in a complex environment like Google Research Football is a good choice, although due to issues with Google Research Football (such as the exploitability of the built-in AI) it is perhaps not as complex as the authors may hope, even though it has presented a challenge to past MARL research. However, this is a broader issue within the MARL community and the authors of this paper cannot fairly be singled out for this.

---

> ### Author Rebuttal · Authors · 2023-08-09
>
> Dear Reviewer k74L,
>
> Thanks for your review of our paper. Here's our response to the weaknesses, questions, and limitations you have highlighted:
>
> -----
>
> **Q1:** computational resources
>
> **A:** IPPO is directly trained in GRF 5v5, while SPC or other curriculum learning baselines are trained in training tasks. The computational resources are the same for these baselines since the computational overhead mainly depends on the inference/backpropagation of the neural network, which is aligned in this implementation. We implemented SPC and VACL based on RLLib in https://anonymous.4open.science/r/MARL_SPC/.
>
> -----
>
> **Q2:** Experimental setting
>
> **A:** First, the experiments on sparse-reward MPE indicate the MPE environment is relatively simple so curriculum learning is not necessary. Then the GRF checkpoint reward is activated only once for each agent in one single game of training tasks. This reward can only be received once when an agent reaches one checkpoint. The full court is divided into 10 equal intervals, and the endpoints of these intervals are checkpoints.
>
> -----
>
> **Q3:** MADDPG/MAPPO
>
> **A:** Thanks for pointing this out. We will revise our statement in terms of the input of the critic with padding observation.
>
> -----
>
> **Q4:** It's not clear what skills the agents learn in the training tasks that are useful in the target task.
>
> **A:** Thanks for pointing this out. We will add the training task performance throughout the training of SPC in the revised version.
>
> -----
>
> **Q5:** IPPO performance drops
>
> **A:** Thanks for pointing this out. The performance drops around 80 and 90 million steps might be caused by the collapse of training. We will include more experiments with random seeds and smoothen the graph to fix it.
>
> -----
>
> **Q6:** QMix
>
> **A:** We use the RLLib version’s QMix, which will collapse w.r.t. the goal performance to a worse average goal difference than -5.
>
> -----
>
> **Q7:** What is the target distribution for the MPE tasks?
>
> **A:** The target MPE task is cooperative navigation with 16 agents.
>
> -----
>
> **Q8:** The proportions of 3vs1 and Empty-Goal tasks gradually drop.
>
> **A:** The area of  3vs1 and Empty-Goal becomes smaller around 150M steps.
>
> -----
>
> **Q9:** Over-engineered.
>
> **A:** Thanks for your valuable comments on the current presentation. We compared different teacher/auto-curricula algorithms on GRF Corner-5 in Appendix D. The training task space consists of n agents, where n ∈ {1, 3, 5}. All teachers have the same base architecture without transformer architecture and HRL.
>
> Since we aim to solve the challenging cooperative MARL problems, we tested a few methods and found that only one method cannot be the silver bullet. So, we propose SPC, an auto-curricula MARL framework.
>
> As we mentioned in Sec. 3, there are three challenges in designing an auto-curricula MARL algorithm with varying numbers of agents: (1) a non-stationarity curriculum selection problem due to the ever-changing student's strategies; (2) a lack of a general student framework to deal with the varying number of agents; (3) the forgetting and relearning problem. The three issues are related to the curriculum setting and are not easy to be decoupled in this paper setting.
>
> Towards these challenges, we first describe our curriculum learning algorithm in Sec 3.2 with detailed design and regret analysis. And then describe the hierarchical structure and communication structure in fewer words in Sec 3.3. Note that the student policy architecture introduced in this paper is fixed and not changed when the training tasks (with different numbers of agents) are changing.
>
> We admit that the hierarchical/communication structure is less novel. However, proposing new HRL/communication algorithms is not the core idea of the paper. We just show one possible solution to deal with the mentioned challenges. The hierarchical/communication compositions could have different algorithms, which can be used in SPC.
>
> -----
>
> **Q10:** The shooting reward
>
> **A:** If without the shooting reward, only checkpoints reward and goal reward will lead to a running behavior.

---

### Official Review · Reviewer_oZVt · 2023-07-08

**Soundness:** 3 good
**Presentation:** 2 fair
**Contribution:** 3 good
**Rating:** 7
**Confidence:** 4

**Summary:**

The paper introduces a new automatic curriculum learning framework, Skilled Population Curriculum (SPC), for multi-agent reinforcement learning. The algorithm includes three major components: (1) a contextual bandit conditioned by student-policies representation for automatic curriculum learning; (2) An attention-based communication architecture for policies to learn cooperation and behavior skills from distinct tasks with varying numbers of agents; (3) A hierarchical policy architecture to help agents to learn transferable skills between different tasks. The experiments are conducted in Google Research Football environment and Multi-agent Particle environments, which demonstrate the efficiency of the proposed method to IPPO and VACL.

**Strengths:**

1. The proposed method is simple yet efficient in the complex Google Research Football environment.
2. The motivation of the components are also clear and make sence.
3. In exepriment, several ablation studies demonstrate the effectiveness of the proposed components;
4. Also, the paper is overall easy to follow to me. The key idea is easy to understand.

**Weaknesses:**

This paper could benefit from further improvements in the following aspects:

1. It seems that the manuscript introduces various components. While each one appears to be intuitive and rational in isolation, I recommend that the authors should provide a unifying theme or framework to better connect these components. Presently, it appears as if these components are addressing three discrete issues: a) efficient curriculum learning, b) policy architecture development, and c) communication in varying agent scenarios. It is noteworthy that a paper does not necessarily need to devote substantial attention to the innovative aspects of each introduced components. In the case of this paper, the hierarchical structure, for instance, appears to be a standard approach with limited novelty. The authors can highlight how they design efficient automatic curriculum learning in the context of variable agent scenarios.

2. In the section discussing related work (Line 221), the authors mention various curriculum learning mechanisms without a detailed discussion. Could the authors provide an expanded explanation on how these works conduct curriculum learning and how they relate to or differ from the proposed methodology?

3. There is room for improvement in the experiments section. Specific recommendations are detailed in the questions section.

4. The paper could be further polished, for instance:
    - There are several instances where a capital letter follows a comma, such as in line 40: "For example, In the football environment, when we…"
    - The legend of Figure 6(b) lacks clarity. It would be beneficial if the authors could provide a detailed explanation of what the labels 0,1,2,3 represent.

**Questions:**

1. Could the authors clarify why maximizing rewards should be the objective of a teacher in curriculum learning? Intuitively, it seems a teacher policy that aims to maximize performance given student policies would tend to recommend simpler tasks to learn, which is not what we would like to see.

2. Ignoring the previous question, could the authors explain why the Bandit model necessitates using the representation of students' policies as input of the teacher policy? It seems that providing an optimal course distribution based on the current student's behavior would be sufficient. Why is there a need to consider course distributions under the representations of other (mainly comes from the historical) student policies?

Regarding the experimental section:
1. At Line 293, the authors mention that the SPC can switch to the largest population rapidly. Could the authors provide further explanation as to why this is possible and why it represents an advantage?
2. The second and third paragraphs of Section 5.3 are unclear; the authors could rephrase them for clarity. You could try presenting your point as follows: "From figure X (or the comparison of X and Y), we can observe XX, which indicates XX."
3. In Appendix C, a more challenging 11 vs. 11 experiment was introduced, with the authors claiming superior performance of the SPC, which is great to see. But this claim raises questions as there are no baselines for comparison. Could the authors consider adding some baselines to this task?

**Limitations:**

NAN

---

> ### Author Rebuttal · Authors · 2023-08-09
>
> Dear Reviewer oZVt,
>
> Thanks for your review of our paper. Here's our response to the weaknesses, questions, and limitations you have highlighted:
>
> -----
>
> **W1:** It seems that the manuscript introduces various components.
>
> **A:** Thanks for your valuable comments on the current presentation.  Please see joint reponse.
>
> -----
>
> **W2:** Could the authors provide an expanded explanation on how these works conduct curriculum learning and how they relate to or differ from the proposed methodology?
>
> **A:** ADR [31, 32] is proposed by OpenAI to solve Rubik's cube by realizing a training curriculum that gradually increases the difficulty. The additional training environments are only added when a minimum level of performance is achieved. Different from ADR, SPC automatically selects new tasks instead of using a threshold.
>
> ALP-GMM [33] fits a Gaussian Mixture Model (GMM) as the teacher on a dataset of Absolute Learning Progress (ALP) measure, where ALP = | r_new - r_old|, r_new, r_old are mean episode rewards under new or old training task distribution. ALP-GMM aims to maximize average competence over a given parameter space. Different from ALP-GMM, SPC uses mean episode rewards under testing task distribution as Absolute Learning Progress.
>
> SPCL [34] is about machine learning instead of reinforcement learning, which embeds curriculum design as a regularization term into the learning objective. SPDL [38] utilized SPCL by introducing a KL divergence regularization term between training probability distributions and testing probability distributions. CURROT [39] utilized SPCL by introducing Wasserstein distances regularization terms between training probability distributions and testing probability distributions. Different from these methods, SPC model curriculum learning as a two-level optimization instead of introducing a regularization term.
>
> GoalGAN [35] uses GAN to generate a new curriculum. Different from GoalGAN, SPC uses a multi-arm bandit algorithm instead of a GAN for stable training.
>
> PLR [36, 37] selectively samples the next training level given the current policy, by prioritizing levels with higher estimated learning potential when replayed. Different from PLR, SPC uses on-policy RL algorithms as the backbone instead of off-policy RL algorithms with replay buffer which might lead to the forgetting and relearning problem.
>
> Graph-curriculum [40]  introduces a heuristic to guide curriculum learning algorithm. The heuristic assumes the larger the size of the state space is, the harder the training task is. Different from Graph-curriculum, SPC automatically selects new tasks instead of using manual configuration.
>
> Different from all these methods, SPC aims to solve the challenging cooperative MARL problems by designing an auto-curricula MARL algorithm with varying numbers of agents. While these methods focus on single-agent RL problems.
>
> We will include this discussion in the Appendix.
>
> -----
>
> **W3:** The paper could be further polished.
>
> **A:** Thanks for your valuable comments on the current presentation. The labels 0,1,2,3 represent a default kill and skill 1,2,3 in Appendix D. We will fix these typos and presentations.
>
> -----
>
> **Q1:** Could the authors clarify why maximizing rewards?
>
> **A:** Intuitively, the objective of a teacher is to make the student better. It is measured by the mean episode reward of the student in the testing environment in SPC. SPC selects different training environments for students to learn skills based on their performance in testing environments. Even though the students get higher rewards in simpler training tasks, it doesn’t mean that the skills learned in training tasks can lead to a higher score in testing. For example, players learn to run well in the most simpler task but cannot shoot in 5v5 competition. The measurement in SPC is for handling the forgetting and relearning problem.
>
> **Q2:** Could the authors explain why the Bandit model necessitates using the representation of students' policies as input of the teacher policy?
>
> **A:** Since the curriculum learning from the perspective of the teacher is non-stationary. That is, given the same arm selection, different students learning status leads to different rewards for the teacher. For example, a rookie and an expert will perform differently under the same training distribution. It necessitates the multi-armed bandit algorithms to consider the student learning status as context. So, SPC uses RNN to capture the historical student's behavior to approximate the students learning status.
>
>
> -----
>
> **Q3:** Could the authors provide further explanation as to why this is possible and why it represents an advantage?
>
> **A:** Since IPPO is trained and evaluated directly on the target task, it can achieve a not-bad performance. It indicates that the MPE environment is relatively simple so curriculum learning is not much necessary. So it is an advantage of SPC that it can switch to the largest population rapidly to train on the most helpful tasks instead of staying simple tasks.
>
> -----
>
> **Q4:** The second and third paragraphs of Section 5.3 are unclear; the authors could rephrase them for clarity.
>
> **A:** Thanks for your comments on the presentation of Sec 5.3. We will fix these presentations.
>
> -----
>
> **Q5:** In Appendix C, a more challenging 11 vs. 11 experiment was introduced, with the authors claiming superior performance of the SPC, which is great to see. But this claim raises questions as there are no baselines for comparison. Could the authors consider adding some baselines to this task?
>
> **A:** GRF 11vs11 is super hard for current MARL algorithms and to our best knowledge no algorithm is reported to handle the GRF 11vs11. We tested other algorithms but it requires much computation resources which is out of our capacity. However, we opensource our code  https://anonymous.4open.science/r/MARL_SPC/ for the community for further research.

---

> > ### Comment · Reviewer_oZVt · 2023-08-19
> >
> > Thanks for the response. Parts of my concerns have been solved. I have the following-up questions.
> >
> >
> > For previous Q1: The author says the student's fitness values are estimated in a test environment, but why can this be the preference of the teacher policy to improve learning efficiency? Also, as mentioned by the authors, players learn to run well in the most simpler task but cannot shoot in 5v5 competition, so how can your student policies complete the tasks in your test environment?
> >
> > For previous  Q2: I known that given the same arm selection, different students learning status leads to different rewards for the teacher,  but why do we need to consider the students learning status? I think providing an optimal course distribution based on the **newest** student's behavior would be sufficient to guide the student learning. Do I miss something?
> >
> > For previous Q3: so why can SPC switch to the largest population rapidly?

---

> > > ### Author Response · Authors · 2023-08-20
> > >
> > > We appreciate your comments and effort. Below is our answers to your new questions:
> > >
> > > -----
> > >
> > > **Q1:** Student's fitness values as the preference of the teacher policy
> > >
> > > **A:** This concept is from curriculum learning. As shown in our related work, most curriculum learning methods use "Learning Progress" as a metric to determine whether a curriculum selection is good. Different methods have different formulations of "Learning Progress". For example, ALP-GMM [33] uses ``ALP = |r_new - r_old|``, where r_new, and r_old are mean episode rewards under new or old training task distribution. VACL uses the difference of value function ``V_new(s)-V_old(s)``, where $s$ is sampled from the state distribution in testing task $p_{test}(s)$. In SPC, we use the testing return as "Learning Progress", that is, $return_{5v5}(\pi_{new student})$.
> > >
> > > Players learn to run well in the most simpler task but cannot shoot in 5v5 competition, so the reward in early learning is low. Bandit teacher would try other arms/training tasks, such as a 3v1 shooting task. In this case, once the players learn to shoot, they can get higher rewards in 5v5. So the testing reward can be the measure of how good the curriculum selection is.
> > >
> > > -----
> > >
> > > **Q2:**  Why do we need to consider the students learning status?
> > >
> > > **A:** The student's learning status is exactly indicating the newest student's behavior. The reason why we consider the students learning status is the forgetting and relearning problem. In this case, the students are not always improving themselves. We should always provide a course for students. Our bandit teacher would exploit and explore the course distribution. (Other curriculum learning methods are also based on such perspective of course distribution.) In the teacher's exploration, the students might learn easier or unrelated tasks, leading to forgetting learned skills. The player might go back to the previous learning status. For example, if a coach already taught a player how to make a three-point shot, then this coach teaches the player how to make a slam dunk. Then the player might forget how to make a three-point shot well. So the teacher should take the current student's learning status into consideration.
> > >
> > > -----
> > >
> > > **Q3:** Why can SPC switch to the largest population rapidly?
> > >
> > > **A:** SPC can switch to the largest population rapidly in the MPE environment. During the exploitation and exploration of the curriculum selection, the teacher notices that providing tasks with a larger population can lead to a larger reward, SPC can switch to the largest population rapidly.
> > >
> > > -----
> > >
> > > We thank you for your endorsement of the motivation and the experimental studies of our work We appreciate your positive comments on the quality and experimental studies of our work. If you have any further questions or comments, we will be happy to discuss or fix them further. We are looking forward to your feedback.

---

> > > > ### Author Response · Authors · 2023-08-21
> > > > **Further Clarification**
> > > >
> > > > We would like to clarify our method more.
> > > >
> > > > -----
> > > >
> > > > **Q:** Why testing returns of the student are set as the reward for the teacher?
> > > >
> > > > **A:** From the perspective of the bandit teacher, it will decide to select an arm or a training task and then get a reward after student training. Such a reward should reflect how good the task selection is. A good training task should help the student to get a high score on the testing task. So we directly use the testing returns of the student as the reward for the teacher.
> > > >
> > > > -----
> > > >
> > > > **Q:** Why do we need context?
> > > >
> > > > **A:** The teacher is facing a non-stationary environment. The same action of the teacher will lead to different rewards w.r.t. different student learning statuses. We use an RNN to capture student learning status as context for the teacher to alleviate the non-stationarity.
> > > >
> > > > -----
> > > >
> > > > **Q:** The structure of the student module
> > > >
> > > > **A:** SPC consists of a teacher-student structure. In the student structure, we utilize IPPO + HRL + communication to train multiple agents, and these agents maintain their own high-level policies and share one low-level policy. We didn't use any sub-strategies inside the student structure.

---

> > > > > ### Comment · Reviewer_oZVt · 2023-08-22
> > > > >
> > > > > The authors have addressed my major concerns. Thank for the response.

---

### Official Review · Reviewer_p3Bh · 2023-07-09

**Soundness:** 2 fair
**Presentation:** 3 good
**Contribution:** 2 fair
**Rating:** 5
**Confidence:** 3

**Summary:**

This work introduces the Skilled Population Curriculum (SPC), an automated curriculum learning algorithm designed for Curriculum-enhanced Dec-POMDP. The goal of SPC is to enhance the student's performance on target tasks via a sequence of training tasks provided by the teacher. The SPC functions as a nested-HRL method, where the teacher serves as the upper-level policy and is modeled as a contextual multi-armed bandit. At each teacher timestep, the teacher selects a training task from the distribution of bandit actions, with the context derived from the student policy's hidden state. The teacher's bandit is optimized using the student policy's test reward. The lower-level policy, also known as the "student", is in itself a hierarchical policy. The high-level policy implements population-invariant communication using a self-attention communication channel to manage messages from a number of agents, and all students share the same low-level policy.

**Strengths:**

- This paper is well-presented. Figure 1 is well-designed. I can get a good understanding of this paper's method just by reading this figure.
- The algorithm is implemented with Ray RLlib, though the code is not currently available.

**Weaknesses:**

- **This study seems to be an overcomplicated amalgamation of pre-existing methods.** SPC stacks three layers of hierarchical policies (teacher 1 + student 2), the teacher is modeled as a multi-arm bandit with a fixed output dimension (number of tasks), and the lower-level control policies of the students are shared. The intricacy of this pipeline leads me to question its generalizability and practical applicability.
- **More rigorous comparison with current MARL algorithms, and need benchmark results on SMAC**, which is de facto the most standard benchmark for MARL algorithms. Please consider adding [MAPPO](https://github.com/marlbenchmark/on-policy), [HARL](https://github.com/PKU-MARL/HARL), and their multi-agent communication variant as your baselines.
- Line 236-238, “However, current approaches that extend HRL to multi-agent systems or utilize communication are limited to a fixed number of agents and lack the ability to transfer to different agent counts”, this is an inaccurate claim because it has been done in the ICLR 2022 publication, [*ToM2C*](https://arxiv.org/pdf/2111.09189.pdf), which similarly uses the HRL with a population-invariant multi-agent communication mechanism. AFAIK this cannot be treated as "communication limited to a fixed number of agents". Please consider citing this work and changing your statement regarding the previous work.

**Questions:**

- **I need more convincing results for proving "The Necessity of Curriculum Learning".** Why is an in-depth analysis of the teacher-student framework necessary? Despite its significantly increased implementation complexity compared to the original MAPPO algorithm, their performances appear roughly equivalent. Moreover, the MAPPO algorithm, provided sufficient exploration and large batch size, has already demonstrated state-of-the-art performance on both SMAC and GFR benchmarks. I presume an advantage of combining the ACL and the teacher-student framework lies in handling more challenging scenarios through incremental learning. To underscore the superiority of SPC over traditional multi-agent PPO methods, could you present performance data from the SMAC Super-Hard Map difficulty? This could include instances like 3s5z_vs_3s6z, where MAPPO previously underperformed significantly.
- I am interested in understanding the implementation of multi-agent communication in Ray RLlib. It appears that the agents are exchanging messages before outputting their current actions. It's somewhat challenging for me to envision how this process is technically executed within the RLlib framework. I wish the code is available.

**Limitations:**

The limitations of this paper are only briefly mentioned in the last section.

---

> ### Author Rebuttal · Authors · 2023-08-09
>
> Dear Reviewer p3Bh,
>
> Thanks for your review of our paper. Here's our response to the weaknesses, questions, and limitations you have highlighted:
>
> -----
>
> **W1:** Complexity and Generalizability.
>
> **A:** While we acknowledge that our study introduces an intricate pipeline, this approach has been adopted to address certain challenges that conventional methods fail to accommodate. Since we aim to solve the challenging cooperative MARL problems, we tested a few methods and found that only one method cannot be the silver bullet. As we mentioned in Sec. 3, there are three challenges in designing an auto-curricula MARL algorithm with varying numbers of agents: (1) a non-stationarity curriculum selection problem due to the ever-changing student's strategies; (2) a lack of a general student framework to deal with the varying number of agents; (3) the forgetting and relearning problem. The three issues are related to the curriculum setting and are not easy to be decoupled in this paper setting.
> Furthermore, the proposed hierarchical policies are designed to introduce structured learning, which in turn, we believe, enhances the method's generalizability. We also show SPC’s performance in 11vs11 in the appendix to showcase its practical applicability.
>
> -----
>
> **W2:** Comparison with Other MARL Algorithms.
>
> **A:** The experiments demonstrate the effectiveness of each module of SPC. We choose IPPO for comparison because we also use IPPO as SPC’s backbone. We didn’t include other MARL algorithms since the curriculum is orthogonal to the backbone policy optimization algorithms such as MAPPO and HARL. Such MARL can be easily extended by introducing a teacher module and an RNN module to record historical behavior.
>
> -----
>
> **W3:**  Please consider citing ToM2C and changing your statement regarding the previous work.
>
> **A:** Thank you for pointing out the ToM2C paper. We'll incorporate this reference and modify our statement to give due credit to the prior work.
>
> -----
>
> **Q1:** On the Necessity of Curriculum Learning. Why is an in-depth analysis of the teacher-student framework necessary?
>
> **A:** Experimentally, we tested a few methods and found that only one method cannot be the silver bullet. We compared different teacher/auto-curricula algorithms on GRF Corner-5 in Appendix D. The training task space consists of n agents, where n ∈ {1, 3, 5}. All teachers have the same base architecture without transformer architecture and HRL. We can see that without curriculum learning algorithms, the performance of “None” is not competitive. The in-depth analysis of the teacher-student framework is primarily to elucidate the reasons we introduce each module in SPC. Curriculum learning is beneficial when solving complex tasks in many domains. With the analysis, we point out the drawbacks of current curriculum learning methods when applied in MARL settings.
>
> -----
>
> **Q:** On the SMAC benchmark.
>
> **A:** SMAC environment is a battle-based environment. The agents are encouraged to attack enemies one by one. In this case, the agents are supposed to have same behavior. SPC aims to learn complex cooperation with sparse reward in MARL. SOTA algorithms already can learn the hardest scenarios in SMAC without curriculum, while cannot learn in GRF 5v5. For example, RODE has already achieved median win rate of 96.8 in the 3s5z_vs_3s6z you mentioned [1,2]. Therefore, we did not include SMAC.
>
> -----
>
> **Q:** Implementation of multi-agent communication in Ray RLlib.
>
> **A:** We implement multi-agent communication by modifying the single-agent pipeline of RLlib. To achieve this, we update the original PPOTorchPolicy into PPOComTorchPolicy by overriding the computation of loss and GAE to separate the loss for different agents. We also customize a MultiActionDistribution to handle the actions of each agent. For more details, the code for reproduction is available at https://anonymous.4open.science/r/MARL_SPC/.
>
> -----
>
> [1] The Surprising Effectiveness of PPO in Cooperative Multi-Agent Games.
>
> [2] RODE.
>
> -----
>
> We thank you for your endorsement of the motivation and the experimental studies of our work We appreciate your positive comments on the quality and experimental studies of our work. If you have any further questions or comments, we will be happy to discuss or fix them further. We are looking forward to your feedback.

---

> > ### Comment · Reviewer_p3Bh · 2023-08-16
> > **Thanks**
> >
> > Thanks for your rebuttal. I can see the amount of effort spent, and it addresses most of my questions, hereby raising my score. Please consider uploading the updated version of your manuscript asap.

---

### Official Review · Reviewer_reFd · 2023-07-31

**Soundness:** 2 fair
**Presentation:** 3 good
**Contribution:** 2 fair
**Rating:** 5
**Confidence:** 4

**Summary:**

This paper studies the multi-agent RL problem with sparse reward and a varying number of agents. The authors propose a novel automatic curriculum learning strategy to solve complex cooperation tasks in this setting. Their curriculum strategy involves a teacher component and a student component. The teacher component selects the sequence of training tasks for the student component using the contextual bandit algorithm with predictive representation of the student’s current policy as context. The student component is endowed with a hierarchical skill framework and population-invariant communication. They empirically investigate their proposed strategy in two environments (MPE and GRF).

**Strengths:**

The paper is overall well-written, and the related work is extensively discussed.

The theoretical results in this paper seem correct; I haven’t checked the details of the proofs.

The population invariant communication module is an interesting contribution to dealing with the varying number of agents across tasks. It would be interesting to compare its effectiveness (on its own) against the existing methods to deal with varying numbers of agents [23, 24].

**Weaknesses:**

I am unsure about the broader applicability of the contextual representation of the student policy using an online clustering algorithm. How much information will be lost in this process for a high-dimensional policy (e.g., that operates on image inputs)?

Presented experimental results are not sufficient to validate the effectiveness of the proposed curriculum strategy (specifically the teacher component) in complex scenarios, given that in the MPE environment, the impact/necessity of curriculum is negligible.

**Questions:**

Why EPC [9] is not used as a baseline in the experiments?

The authors mention, "To ensure a fair comparison, we modify VACL by removing the centralized critic for MPE tasks.” Please explain why.

In Figure 5, including the results for SPC w/o “both” HRL and COM would be good. Then, we can see the effectiveness of the teacher component (or curriculum).

In Figure 3 (MPE environment), including the ablation study results (similar to Figure 5) would be good.

It is also important to discuss/report the proposed strategy's computational cost/overhead (run time) compared to the baselines like VACL.

**Limitations:**

The paper is of an algorithmic nature and does not have any direct potential negative societal impact.

---

> ### Author Rebuttal · Authors · 2023-08-09
>
> Dera Reviewer reFd,
>
> Thanks for your review of our paper. Here's our response to the weaknesses, questions, and limitations you have highlighted:
>
> -----
>
> **W1:** I am unsure about the broader applicability of the contextual representation of the student policy using an online clustering algorithm. How much information will be lost in this process for a high-dimensional policy (e.g., that operates on image inputs)?
>
> **A:** An **online** clustering algorithm is used to classify the **ever-changing** student policy (which is represented by a context vector) in an end-to-end manner. The context vector indicates the learning status of the students. The clustering operation means discretizing the context vector to satisfy the regret analysis in Sec 3.2.2. The information loss depends on the number of cluster centers, which is also the number of arms in multi-arm bandit algorithms. Based on Theorem 3.4, the more cluster centers are, the larger regret bound becomes.
>
> -----
>
> **W2:** Presented experimental results are not sufficient to validate the effectiveness of the proposed curriculum strategy (specifically the teacher component) in complex scenarios, given that in the MPE environment, the impact/necessity of curriculum is negligible.
>
> **A:** We mainly focus on GRF since there are some challenges in the GRF. (1) Large-scale problem: In the GRF, for cooperative players, the joint action space is large; therefore, it is difficult to build a single agent to control all players. Moreover, the opponents are not fixed due to a stochastic environment and a difficult configuration, and the agents should be adapted to various opponents. (2) Sparse rewards: The goal of the football game is to maximize the scores, which can only be obtained after a long time by iteration.
>
> Other environments like the SMAC benchmark are not studied here since the SMAC environment is a battle-based environment. The agents are encouraged to attack enemies one by one. In this case, the agents are supposed to have the same behavior. SPC aims to learn complex cooperation with sparse reward in MARL. SOTA algorithms already can learn the hardest scenarios in SMAC without curriculum, while cannot learn in GRF 5v5. So we did not include SMAC.
>
> -----
>
> **Q1:** Why EPC [9] is not used as a baseline in the experiments?
>
> **A:** We didn’t compare with EPC for three reasons.
> 1. EPC increases the number of agents by twice every curriculum update, which doesn’t support the arbitrary number of agents.
> 2. EPC doubles the number of agents by **cloning** each of the existing agents, which leads to similar behavior of agents.
> 3. The core idea of EPC is to progressively increase the population of agents throughout the training process for large-scale settings. While SPC is proposed to solving different tasks with varying numbers of agents.
>
> -----
>
> **Q2:** The authors mention, "To ensure a fair comparison, we modify VACL by removing the centralized critic for MPE tasks.” Please explain why.
>
> **A:**We remove the centralized critic in VACL since SPC uses independent PPO without the centralized critic as the backbone. We aligned the actor-critic structure of SPC and VACL. Maybe an argument would be made that the use of a transformer architecture in SPC can be effectively viewed as centralized training since the gradient is passed across agents during training. In this paper, we remove the communication and compare SPC without gradient-pass across agents and VACL in Fig. 5 for a fair comparison.
>
> -----
>
> **Q3:** In Figure 5, including the results for SPC w/o “both” HRL and COM would be good. Then, we can see the effectiveness of the teacher component (or curriculum).
>
> **A:** SPC = bandit teacher + IPPO with communication and HRL. So SPC w/o “both” HRL and COM is only bandit teacher + IPPO. We add the new experiments SPC w/o “both” HRL and COM. It achieves a 0.53±0.07 win-rate in GRF 5v5.
>
> -----
>
> **Q4:** In Figure 3 (MPE environment), including the ablation study results (similar to Figure 5) would be good.
>
> **A:** We conducted the ablation study by removing components of SPC and will update Fig.3 in a revised version.
>
> -----
>
> **Q5:** It is also important to discuss/report the proposed strategy's computational cost/overhead (run time) compared to the baselines like VACL.
>
> **A:** We implemented SPC and VACL based on RLLib in https://anonymous.4open.science/r/MARL_SPC/. The running time of SPC and VACL is similar since the computational overhead mainly depends on the inference/backpropagation of the neural network, which is aligned in this implementation.

---

> > ### Comment · Reviewer_reFd · 2023-08-21
> >
> > I thank the authors for their response and for addressing my concerns.

---

### Author Rebuttal · Authors · 2023-08-09

We thank you for your endorsement of the motivation and the experimental studies of our work We appreciate your positive comments on the quality and experimental studies of our work. If you have any further questions or comments, we will be happy to discuss or fix them further. We are looking forward to your feedback. Here we provide the joint question about this work.

-----

### Q: Complexity and Generalizability

A: Since we aim to solve the challenging cooperative MARL problems, we tested a few methods and found that only one method cannot be the silver bullet. So, we propose SPC, an auto-curricula MARL framework.

As we mentioned in Sec. 3, there are three challenges in designing an auto-curricula MARL algorithm with varying numbers of agents: (1) a non-stationarity curriculum selection problem due to the ever-changing student's strategies; (2) a lack of a general student framework to deal with the varying number of agents; (3) the forgetting and relearning problem. The three issues are related to the curriculum setting and are not easy to be decoupled in this paper setting.

Towards these challenges, we first describe our curriculum learning algorithm in Sec 3.2 with detailed design and regret analysis. And then describe the hierarchical structure and communication structure in fewer words in Sec 3.3. Note that the student policy architecture introduced in this paper is fixed and not changed when the training tasks (with different numbers of agents) are changing.

We admit that the hierarchical/communication structure is less novel. However, proposing new HRL/communication algorithms is not the core idea of the paper. We just show one possible solution to deal with the mentioned challenges. The hierarchical/communication compositions could have different algorithms, which can be used in SPC.

The contributions of SPC are: (1) utilizing a teacher-student framework to train agents with varying numbers; (2) proposing a multi-armed bandit algorithm to handle the non-stationary non-differentiable joint teacher-student optimization; (3) introducing the hierarchical and communication structure for a general student framework. We will highlight our contributions, especially the design of efficient automatic curriculum learning in the revised version.

---

### Decision · Program_Chairs · 2023-09-21

**Decision:**

Reject

**Comment:**

The paper studies the problem of automatic curriculum design for multi-agent RL in complex,  sparse reward environments. The proposed curriculum approach, Skilled Population Curriculum (SPC), is inspired by team sports where players train by gradually increasing the task difficulty and number of coordinating players. SPC comprises several components, including a teacher module based on contextual bandits, a population-invariant communication framework in the student module to handle varying number of agents, and a hierarchical skill framework in the student module to learn transferable skills. Finally, SPC is evaluated on several multi-agent RL environments. The reviewers acknowledged that the paper investigates an important problem setting of automatic curriculum design in multi-agent RL. However, there was some spread in the reviewers' assessment, and the reviewers raised several concerns in their initial reviews. We want to thank the authors for their hard work preparing the detailed responses and actively engaging with the reviewers during the discussion phase. These responses did help in improving the reviewers' ratings about the paper; however, the paper still stands as borderline. In the follow-up discussions, several reviewers still raised concerns that the proposed approach is quite complex with somewhat limited novelty, and the provided empirical evidence is insufficient to flesh out the utility of the teacher component. Based on these discussions, unfortunately, the final decision is a rejection. Nevertheless, this is potentially impactful work, and we encourage the authors to incorporate the reviewers' feedback when preparing a future paper revision.